# Upstream and Downstream Bioprocessing in Enzyme Technology

**DOI:** 10.3390/pharmaceutics16010038

**Published:** 2023-12-27

**Authors:** Nadia Guajardo, Rodrigo A. Schrebler

**Affiliations:** 1Departamento de Ingeniería Química y Bioprocesos, Escuela de Ingeniería, Pontificia Universidad Católica de Chile, Avenida Vicuña Mackenna 4860, Macul, Santiago 7820436, Chile; 2IONCHEM SpA, El Tordillo, 154, Villa Alemana, Valparaíso 2471548, Chile; rodrigoschrebler@hotmail.com

**Keywords:** upstream processing, downstream processing, enzyme technology, separation processes, bioprocessing

## Abstract

The development of biotransformation must integrate upstream and downstream processes. Upstream bioprocessing will influence downstream bioprocessing. It is essential to consider this because downstream processes can constitute the highest cost in bioprocessing. This review comprehensively overviews the most critical aspects of upstream and downstream bioprocessing in enzymatic biocatalysis. The main upstream processes discussed are enzyme production, enzyme immobilization methodologies, solvent selection, and statistical optimization methodologies. The main downstream processes reviewed in this work are biocatalyst recovery and product separation and purification. The correct selection and combination of upstream and downstream methodologies will allow the development of a sustainable and highly productive system.

## 1. Introduction

The development of a robust bioprocess must include the proper integration of upstream and downstream processes [1,2,3,4].

Upstream bioprocessing comprises tasks in the initial stages of biotransformation [2,5], but these will also affect downstream processes. These stages include the preparation of biocatalysts, the use of solvents as a part of the reaction medium, and optimization methodologies (See Figure 1) [6,7,8,9,10].

A key aspect before starting an enzymatic process is the availability of the enzyme on an industrial scale [11,12,13]. The enzyme market worldwide is growing at 6.8% [14]. Among the most used enzymes are the group of carbohydrases, hydrolases, polymerases, and nucleases [14]. They are used in sectors of industry such as food, nutraceuticals, detergents, animal feed, and biofuels. Most enzymes are produced by microorganisms, followed by those produced by animal sources, and, finally, those produced by plant sources. This information indicates the excellent availability of enzymes in these industrial areas [14].

Enzymes derived from natural sources often exhibit limited activity and stability due to their biological origin. Biocatalyst engineering consists of modifying the enzyme using protein engineering [15,16] through immobilization techniques [17,18] or a combination of these [19]. Protein engineering involves the strategic modification of specific amino acids within the protein’s structure with the primary objective of enhancing the catalytic performance of enzymes [20]. Another strategy to stabilize and reuse the enzyme is immobilization, which consists of fixing the enzyme on a material (binding to support) or forming cross-linked enzyme aggregates (CLEAs) between the same molecules [21,22]. Physical and covalent bonds are among the support binding methods [23,24]. The physical unions are carried out on hydrophobic materials, weak Van der Wals unions, or by encapsulation using polymers, biopolymers, or a combination of both [25]. The covalent binding of enzymes to a support is carried out using a bifunctional chemical compound that binds to the chemical groups of the support and the amino acids on the enzyme’s surface, such as the lysine groups [18]. Mixed methodologies also combine physical and covalent binding, such as heterobifunctional supports, where the enzyme first binds hydrophobically and then covalently [18].

Statistical optimization is a valuable tool for finding the optimal conditions for variables that affect the conversion or yield of a reaction. To obtain suitable adjustments and approximations, it is essential to carry out previous experimental tests and more than one experimental design [26].

Upstream processes in enzymatic biocatalysis are only sometimes necessary since they depend on the requirements of a new technology. For example, some companies buy biocatalysts that are already immobilized while others generate them tailor-made for specific applications and at lower costs [27]. It is crucial to emphasize that in industrial processes where the enzyme is not recycled, these methodologies will not be included in the upstream processes.

Downstream processing ensures the efficient recovery of commercially viable products. These processes in enzymatic systems consist of biocatalyst recovery (if it is immobilized) and separation and purification of the reaction products [28,29] (See Figure 1). As shown in Figure 1, downstream processes are often more numerous than upstream processes, which is why, as mentioned above, they are more expensive [30]. In some cases where high-purity end products are required, the cost can reach 80% of the total cost of the production process [31,32]. Likewise, the need to purify products of interest in downstream processes will depend on their application. For example, if the final product is a pharmaceutical compound, it must have a high purity, like food industry products.

In contrast, some bulk compounds do not require much purification [28]. Regarding the recovery of the biocatalyst, this review includes centrifugation and filtration processes, and examples of separation and purification that include distillation processes, liquid–liquid extraction, crystallization, and chromatography will be analyzed. In this way, understanding the product’s applications of interest, upstream and downstream strategies in biotransformations can be designed. Given this background, this review aims to scrutinize the latest advances in upstream and downstream processes combined into biotransformations.

## 2. Upstream Bioprocessing

In enzymatic reactions, upstream processes consist of the expression of the enzymes of interest in the cells for their culture, immobilization methodologies for the recovery and stabilization of the biocatalyst, and selection of solvents as part of the reaction medium.

### 2.1. Enzyme Production

Determining a robust biocatalyst to apply as a process catalyst is a challenge due to its intrinsic lability and the need for large-scale enzyme production. For adequate availability of some enzymes, it is essential to follow a series of steps to optimize the production of the enzyme. Initially, it is necessary to isolate the microorganism responsible for enzyme production from a natural source. Once isolated, knowing its genome becomes crucial, as this allows for genetic modifications resulting in a more active and stable enzyme [33,34,35] (Figure 2). These genetic modifications can then be expressed through cloning in microorganisms that are generally recognized as safe (GRAS). In the literature, one can discover many examples showcasing how genetic modification and expression enhance the characteristics and capabilities of enzymes. One of them is the expression of the Lysozyma enzyme from the soil microorganism *Bacillus licheniformis* TIB320 in *Bacillus subtilis*. This enzyme demonstrates high activity and stability in a pH range between 3 and 9 and from 20 °C to 60 °C [36].

According to the existing literature, the enzymes of utmost significance in biocatalysis include lipases, laccases, cellulases, glucosidases, and amylases [37,38].

Research on lipase production has focused on optimizing culture conditions to yield extracellular lipase from *Yarrowia lipolytica* NNRL Y-1095, with a particular emphasis on using a magnetic field (MF). The successful conditions included using 10 g/L of glucose, 15 g/L of olive oil, and 2 g/L of Triton X-100 as surfactants. The highest recorded lipase activity, reaching 34.8 U/mL, was achieved after 144 h of cultivation when applying a 30 mT MF from the 72nd to the 144th hour of the culture period. These findings demonstrate the viability of MF as an alternative approach for the efficient production of lipase from *Yarrowia lipolytica* NNRL Y-1095 [39]; in a subsequent investigation involving the same enzyme production, a biphasic system was established, utilizing tetrabutylammonium bromide ([N_4444_]Br) for lipase separation. This system achieved extraction efficiencies of 100% [40].

The diverse applications of the laccase enzyme have spurred extensive research into enhancing its production while simultaneously reducing production costs. There are two primary strategies for laccase production: submerged fermentation (SF) and solid-state fermentation (SSF). SSF stands out as the superior option due to its high yields and lack of liquid waste generation compared to SF. In solid-state fermentation (SSF), a thin film tightly adheres to the solid substrate; the primary mobile phase is gas, thereby impeding the efficient transfer of mass and heat. Additionally, SSF often involves highly heterogeneous components, further complicating the fermentation process. Addressing these challenges is crucial for advancing industrial fermentation technologies. One of the alternatives proposed by Wang et al. 2023 [41] is pretreatment with sodium hydroxide of lignocellulosic waste that serves as a fixed support, nutrient, and inducer to produce laccase. The results indicate that pretreatment with sodium hydroxide effectively yielded a substrate with significantly enhanced enzymatic digestibility and optimal water retention, thereby creating an ideal environment for promoting the growth of *Funalia trogii* IFP0027 mycelium. Furthermore, the laccase activity recorded reached an impressive 2912.34 U/g, a remarkable 7.72-fold increase compared to the control group [41]. Another investigation points to applying Triton X-100 to obtain a laccase from *Shiraia* sp. S9. It was found that Triton X-100 induces a burst of reactive oxygen species (ROS), increasing membrane permeability, regulating the genetic expression of lactase, and ultimately resulting in higher lactase production, reaching approximately 9.2 × 10^4^ U/L, which is approximately 108 times greater than the control [42].

Cellulases are pivotal enzymes in the enzymatic conversion of complex plant cell-wall polysaccharides into valuable, easily accessible sugars which hold economic importance. Enhancing production performance hinges on a foundational principle: meticulously examining operational variables throughout the fermentation process is essential. In this context, Triphati et al., 2023 [43], studied the effect of temperature and pH on the production of cellulase by SSF. At an optimal substrate concentration (lignocellulosic waste) of 10 g, 22 IU/gds filter paper (FP) was obtained. Optimal conditions for cellulase production were found to be a temperature of 40 °C and a pH of 5.5. These findings underscore the promising feasibility of scaling up industrial cellulase production, utilizing lignocellulosic waste as a cost-effective and sustainable substrate [43]. In another example, the utilization of a substrate combination consisting of sodium carboxymethyl cellulose (CMC-Na) and wheat bran (WB) to produce cellulase from *Penicillium oxalicum* P-07 exhibited remarkable levels of activity on filter paper (FPase). Through a meticulous optimization of degradation conditions, it was observed that at a pH level of 5.0 and a temperature of 37 °C, FPase activity surged by 1.5-fold, escalating from an initial measurement of 0.30 U/mL to an enhanced level of 0.45 U/mL. These finely tuned operational conditions served to promote the efficient production of cellulase by *Penicillium oxalicum* P-07 [44].

β-glucosidase is a versatile enzyme with numerous applications, including the selective hydrolysis of cellobiose to yield fermentable sugars and producing aromatic compounds for use in the food industry. In this context, Oliveira et al., 2021, optimized the carbon source to obtain β-glucosidase from *Malbranchea pulchella* using a central composite experimental design. The model fit coefficient was n (R^2^) 0.9960 with a high F-value. Cellobiose emerged as the optimal carbon source, yielding an enzyme with remarkable performance, boasting a relative activity of 100% and an enzyme activity level of 1.55 U/mg [45]. Another study focused on the production of β-glucosidase by utilizing the microorganism *Aspergillus fumigatus* strain isolated from the Atlantic Forest biome. This enzyme can be efficiently synthesized using an economical culture medium, such as canola meal, as a substrate. Remarkably, the enzyme exhibited a significant activity level, achieving 24.53 U/mL, when cultivated with 2.9% canola flour at a pH of 4.2 over 101 h at a temperature of 40 °C [46].

α-amylase is an enzyme that plays a crucial role in the digestion of carbohydrates in the human body. It is produced in various organs, including the salivary glands and the pancreas. α-amylase breaks down complex carbohydrates, such as starches and glycogen, into simpler sugars like maltose and glucose. Due to its importance, various investigations have aimed to increase its production, such as the research by Abo-Kamer et al., 2023 [47]. The researchers isolated α-amylase from the microorganism Bacillus cereus obtained from soil samples. The isolation process involved serial dilutions and plating techniques, and the amylolytic activity was assessed using the starch agar plate method. Subsequently, we optimized enzyme production through submerged fermentation, employing Plackett–Burman one-factor-at-a-time statistical design. Among the independent variables examined, glucose, peptone, (NH_4_)_2_SO_4_, and MgSO_4_ displayed the highest degree of significance, yielding an impressive 250 U/mL of α-amylase. The physical conditions conducive to optimal enzyme production were a temperature of 35 °C and a pH of 5.5, maintained for 48 h [47].

Another investigation delved into the production of α-amylase from *Pontibacillus* sp. *ZY*. In this study, the focus was on the expression of the enzyme using *Bacillus subtilis* as a host organism, followed by meticulous optimization of both the carbon and nitrogen sources. The noteworthy outcomes revealed that, after fine-tuning the gene expression regulator, signal peptide promoter sequences, and the ribosome binding site (RBS) upstream of the amyZ1 gene, an astonishing enzyme activity of 2.6 times greater than that of the native enzyme was achieved, totaling an impressive 4824.2 U/mL. Moreover, through a rigorous optimization process for the carbon and nitrogen sources, a remarkable α-amylase activity of 49,082.1 U/mL was attained within a 3 L fermenter [37].

### 2.2. Enzyme Immobilization

When it is required to recover and obtain biocatalysts with high stability, the enzyme immobilization strategy is of great importance [48,49,50]. Immobilization consists of fixing the enzyme on a material (binding to support) or forming CLEAs between the same molecules [21,22] (Figure 3).

Physical and covalent bonds are among the support binding methods [23,24]. The physical unions are carried out on hydrophobic materials, weak Van der Wals unions, or by encapsulation using polymers, biopolymers, or a combination of both [25]. Regarding immobilization through adsorption, some of the most frequently employed support materials include methacrylates, silica, magnetic nanoparticles, carbon, and various polymers. An example of this type of immobilization is the adsorption of the enzyme Polygalacturonases from *Piriformospora indica* (Exo-PG) on a new support of κ-carrageenan–chitosan magnetic nanoparticles (CCMNs) and chitosan–pectin magnetic nanoparticles (CPMNs). When Exo-PG was immobilized onto CPMNs and CCMNs, it exhibited impressive specific activities of 479 U/mg and 204 U/mg, respectively. The immobilization of Exo-PG on CPMNs resulted in a remarkable 71.11% recovery of its enzymatic activity accompanied by an impressive immobilization yield of 89.15%. On the other hand, immobilizing Exo-PG on CCMNs led to a slightly lower recovery rate of 65.96% of the enzyme’s activity, with an immobilization yield of 63.85% [51].

An illustrative example involves the immobilization of phospholipase Lecitase Ultra (LU) onto a macroporous resin, specifically LXTE-1000, to catalyze the production of diacylglycerol (DAG) through glycerolysis. The adsorption process lasted 2 h, yielding an impressive 90% immobilization rate. Notably, even after 60 days of storage at both 4 °C and 25 °C, the enzyme retained a remarkable 85% of its activity. The glycerolysis reaction using LXTE-1000-LU resulted in a significant 41.31% yield of DAG [52].

The covalent binding of enzymes to a support is carried out using a bifunctional chemical compound that binds to the chemical groups—such as the lysine groups—of the support and the amino acids on the enzyme’s surface [18]. An example of covalent binding was the immobilization of the enzyme cellulase on functionalized core–shell magnetic gold nanoparticles [53]. The immobilization efficiency by proteins was 84%, and the activity by filter paper (FPase) was 0.78 mmol·mL^−1^. Also, the new biocatalyst was more stable than the free enzyme. *Candida rugosa* lipase has also been covalently immobilized on magnetic Fe_3_O_4_@SiO_2_ [54]. This biocatalyst was used to catalyze the hydrolysis of acidified soybean oil to produce fatty acids. The reaction conditions, such as the amount of biocatalyst, reaction time, and the water/oil ratio, were studied. The optimization results showed that the hydrolysis rate reached 98% with 10 wt% biocatalysts, 3:1 (*v*/*v*) water/oil ratio, and 313 K after 12 h. After five reaction cycles, the hydrolysis activity of the biocatalyst remained at 55%, which is promising for an industrial application [54].

Immobilization by encapsulation of enzymes has also been applied in enzymatic reactions of industrial interest, such as immobilization of *Aspergillus oryzae lipase* (AOL) and *Rizomucor miehiei* (RML) lipase on magnetic chitosan microcapsules. The encapsulation process was carried out by self-assembling negatively charged magnetic nanoparticles (citrate-modified Fe_3_O_4_) on cross-linked aggregates of chitosan citrate lipases [55]. Integrating Tween 80 bio-imprinting of AOL@RML co-immobilized lipases, a biodiesel yield of 98% was achieved, which is 1.3 and 1.6 times higher than that of individually immobilized AOL and RML enzyme [55]. The biocatalyst exhibited remarkable stability, retaining 96% of its initial activity after three consecutive reaction cycles.

There are various examples of the immobilization of enzymes using CLEAs [56]; one immobilization method involves using cross-linked aggregates of *Trametes versicolor* laccase with the assistance of sugars. The optimization process focused on critical variables such as the choice of precipitant, cross-linking duration, type of cross-linking agent, and sugar concentration. These optimized conditions were then compared to laccase-CLEAs enriched with bovine serum albumin (BSA). The results revealed that the fructose-CLEAs immobilization method achieved an impressive recovery of approximately 45% of the enzyme’s activity. Furthermore, the kinetic parameters, specifically the K_cat_ and the K_m_, exhibited values twice as high as those obtained with BSA-CLEAs [57]. An additional example involves the immobilization of *Pseudomonas stutzeri* lipase by forming cross-linked aggregates known as TL-CLEAs. This biocatalyst has demonstrated remarkable resilience and high activity when employed in deep eutectic solvents (DESs). Moreover, it exhibits the potential for reusability for up to three cycles with only a 50% reduction in its efficiency for esterifying benzoic acid with glycerol [22].

Mixed methodologies also combine physical and covalent binding, such as heterobifunctional supports, where the enzyme first binds hydrophobically and then covalently [18]. There are also combinations of union using CLEAs and then a second immobilization by encapsulation using Lentikats^®^ (LentiKat’s a.s., Stráž pod Ralskem, Czech Republic) (Polyvinyl alcohol) to increase stability and mechanical resistance for application in continuous packed bed reactors [58,59].

The selection of the immobilization methodology depends on the type of enzyme, simplicity, cost of the support, and the kind of reactor selected when a process is required to be scaled. For example, in the immobilization of lipases with industrial applications, absorption or heterofunctional methodologies (adsorption and covalent bonding) on methacrylate resins are a good option due to the high mechanical and chemical stability of the support for their application in batch and continuous packed bed bioreactors.

Immobilization techniques by encapsulation or entrapment give rise to biocatalysts with weak mechanical resistance, and their operation in batch bioreactors with mechanical stirrers constitutes an appropriate alternative for their scale-up.

### 2.3. Selection of the Solvent in the Preparation of the Reaction Medium

Applying solvent or co-solvent is crucial for the enzymatic reaction when working with substrates of different polarities. However, its selection is not trivial since it must not affect the enzyme’s activity; it must be environmentally friendly (See Figure 4). In addition to the points noted in Figure 4, the solvent can also improve the enzyme activity to increase the reaction’s conversion and yield. Depending on the polarity of the substrates, green solvents that can dissolve the substrates can be designed. Among green solvents are ionic liquids (ILs) [60,61], DESs [62,63], and bio-based solvents [64,65,66]. ILs are formed by a cation and an anion, are complex to prepare, and are expensive. In contrast, DESs are created by a hydrogen acceptor, such as choline chloride, and a hydrogen donor, which can be alcohols, acids, or urea [67]. They are easy to prepare and cheaper than ILs.

ILs offer several advantages in biocatalysis, primarily, their remarkable substrate solubility. Nonetheless, it is essential to note that the activity and stability of enzymes can be affected when operating within these solvent environments. Cui et al., 2022 [68], determined that hydration of the enzyme is essential for the enzyme to retain its activity and stability. For example, Zhao et al., 2021 [69] synthesized ILs that imitate water with groups such as glycol ether and *tert*-alcohol. These new ILs allowed one to obtain high transesterification activities with lipase B from *Candida antarctica* (Novozym 435^®^, Novozymes, Bagsværd, Denmark) and with the immobilized protease from *Bacillus licheniformis*. Various ILs with water percentages of 2% and 3% *v*/*v* increased the activity of the Novozym 435^®^ biocatalyst to 1.8 times higher than in *tert*-butanol and 1.6 times higher than in di-isopropyl ether [69].

DES has been used for various applications in enzymatic catalysis as a reaction medium, allowing it to carry out reactions with substrates of different polarities [63,70,71]. DES in biocatalysis serves both as solvents and substrates. The addition of a small fraction of water, typically 2–5% *v*/*v*, serves to effectively lower the viscosity of DES, mitigating mass transfer challenges [62,63,72]. In addition, its application has been demonstrated in continuously packed bed bioreactors [62,73] and microbioreactors reaching high yields [72]. Biobased solvents such as CyreneTM have also been used in biocatalysis, mainly as a co-solvent in enzymatic hydrolysis reactions, as a solvent in the esterification of glycerol with benzoic acid [64], and as a co-solvent in the reduction of α-ketoesters [65].

### 2.4. Statistical Optimization

Statistical optimization tools are valuable for obtaining the best operational conditions to conduct the reaction. In the literature, there are numerous investigations of applying this technique to determine the best reaction conditions. For example, enzymatic hydrolysis removes glycerides from acidic rice bran oil (RBAO) [74]. The RBAO comprises fatty acids (FFAs), glycerides, and antioxidants such as γ-oryzanol. The enzymatic hydrolysis was optimized by a face-centered central composite rotatable design to investigate the effects of three independent variables—time, temperature, and water: RBAO ratio—and their interactions on the responses (glyceride removal, γ-oryzanol loss, and FFA production) and to determine the statistical models describing their relationship. Using optimal reaction conditions of 22 h, 48.5 °C, and a water: RBAO ratio of 1:1, a glyceride removal rate of 99%, a loss of γ-oryzanol of 32%, and an FFA production of 73–75% within a prediction interval of 95% were achieved. This work demonstrated that enzymatic hydrolysis is a valuable tool for removing glycerides before extracting the antioxidant γ-oryzanol [74].

Another example of applying statistical optimization is enzymatic hydrolysis via alcalase of *Chlorella microalgae* proteins (Figure 5) [75]. To carry this out, the response surface optimization (RSM) methodology was used. The independent variables studied were pH, reaction temperature, reaction time, and enzyme charge: substrate ratio. The optimal reaction conditions with the enzyme Alcalase were reached at a 3% enzyme load: substrate ratio, 60 °C, pH 6.5 for 3 h of reaction, and a protein concentration of 0.58 mg/mL [75].

Regarding enzymatic synthesis reactions, the optimization of the esterification of 5-Hydroxymethylfurfutal (HMF) with lauric acid has been carried out according to the experimental design of Box–Behnken, using two lipases immobilized (Novozym 435^®^ and *Thermomyces lanuginosus* (TL)) on Immobead 150. With Novozym 435^®^, an optimal conversion of 75% was achieved at 40 °C, 65 mM of HMF, and 16 U Novozym 435^®^. With the TL biocatalyst, a conversion of 78% was reached at 50 °C, 30 mM of HMF, and 16 U [76].

## 3. Downstream Bioprocessing

Advanced recovery, separation, and purification techniques are crucial in developing biocatalytic processes. Downstream operations are essential to maintain the product’s characteristics, aiming at high yield and purity [77,78]. Only the separation and purification of products of enzymatic reactions will be considered in this section.

### 3.1. Filtration

Filtration is a solid–liquid separation technique widely used in biotransformations on a laboratory and industrial scale [79,80]. Filtration is versatile since it can be carried out in continuous and discontinuous modes. Pressure filters use pressures greater than atmospheric, and the pressure differential created across the media causes fluid to flow through the media. In vacuum filters, a slight pressure differential is applied; for this reason, attention must be paid to pressure losses. Low-pressure conditions in a flowing fluid may lead to the phenomenon of “cavitation”, in which dissolved gasses or vapor bubbles are released into the liquid. These filters are widely used in industry because they evaluate a suspension’s filterability and the filter media’s suitability [81].

Galactooligosaccharides (GOSs) are short oligosaccharide chains containing several galactoses and one terminal glucose—the enzymatic synthesis of GOS catalyzed by the enzyme *β*-galactosidase from lactose. Raw GOSs are mixtures containing oligosaccharides, unreacted lactose, and monosaccharides; hence, nanofiltration is an alternative for product purification. Poor membrane selectivity, in terms of separating lactose from GOSs, are their main drawback. For this reason, the purification of enzymatically produced GOSs was carried out by nanofiltration with the incorporation of a previous lactose hydrolysis step. The purification of raw and hydrolyzed GOSs was evaluated using two nanofiltration polymeric membranes (Synder NFG and TriSep XN45) at different transmembrane pressures (8–30 bar) in a stirred dead-end cell. The best operational conditions for hydrolyzed raw GOS nanofiltration were obtained with the TriSep XN45 membrane at 20 bar, 45 °C, and 1500 rpm. Incorporating previous lactose hydrolysis increases GOS retention, monosaccharides, and lactose removal, improving GOS purification [82].

Membrane bioreactors possess the function of improving the biocatalytic reaction and, at the same time, separating the reaction products that are generated. However, despite their exciting applications, their commercial use is restricted due to the system’s catalytic and mass transfer properties. An example in this field is the design of new membranes formed by a co-deposition of polyallylamine hydrochloride/polydopamine (PAH/PDA) to obtain an optimal balance between enzymatic activity, water permeability and enzymatic load (Figure 6) [83]. Flexible immobilization matrices fabricated from fibers possess physical attributes such as a large surface area, light weight, and controllable porosity that grant them the mechanical properties required to create filters, sensors, scaffolds, and other functional interfaces. Coatings can also promote the longevity of biocatalytic membranes, which is why they have attracted interest and application in membrane bioreactors [84].

### 3.2. Liquid–Liquid Extraction

Liquid–liquid extraction is mainly used in biotransformations for the separation of reaction products. Liquid–liquid extraction, or partitioning, is a separation process involving transferring a solute from one solvent to another. The solvents must be immiscible or partially immiscible, and the process is generally carried out by mixing the phases and then separating them [85]. The driving force for the transfer is the difference in the solubility of the target compounds in each phase of the biphasic system.

The extraction can be carried out in batch and continuous mode. Continuous mode is used when the sample volume is large, the distribution constant is unfavorable, or the extraction rate is slow [85]. The solvent selection for the extractions is not trivial. The solvent must possess specific characteristics: it must be immiscible with other solvent, the product of interest must be soluble in one of the phases, and it must be friendly to the environment. This last aspect is essential, and there is much research using ILs and DES as extracting solvents [86,87].

Liquid–liquid extraction with ILs has also been integrated into biocatalytic systems [88,89,90]. ILs are made up of an organic cation and an inorganic anion, and, depending on their components, they are soluble in aqueous and organic media and have a low vapor pressure. However, their high price, toxicity risks, and complicated preparation limit their application in industrial processes [91]. The general approach is to use them as a reaction medium and mix them with the biocatalyst and substrates for the reaction process. Then, the reaction products are extracted with an organic solvent immiscible with ILs, while the biocatalyst and ILs are recycled for the following reaction cycle [92] (Figure 7). The sustainability of this process depends on the price of ILs, ILs recycling, and the type of solvent used for extraction. The most recommended are those derived from renewable sources and include cholinium ethanoate and cholinium propanoate, among others [93].

DESs with similar properties to ILs are easy to prepare; many are biodegradable and formed by a hydrogen acceptor, such as choline chloride, and by a hydrogen donor, such as alcohols, acids, etc. [67,94]. These characteristics make them very attractive for industrial applications. Various examples of DESs are applied in liquid–liquid extraction operations in the literature. One such example is the esterification of ethyl acetate with HMF. In this work, the product of interest was separated by liquid–liquid extraction with a DES formed by choline chloride and glycerol at a molar ratio of 1:2 to achieve a separation efficiency of 81% with only one separation stage [95]. Another work, carried out by González and Guajardo [73], demonstrated that it is possible to integrate a continuous enzymatic esterification process with a semi-continuous liquid–liquid extraction process of the reaction product using a DES formed by choline chloride/glycerol 1/2 (molar ratio) reaching a conversion of 8% and an extraction efficiency of 70% [73]. Pánic et al., 2021 [96], also worked on the kinetic resolution of (R, S)-1-phenyl ethyl acetate using a DES (ChCl: glycerol) as a reaction medium and extractant at the same time. In the final separation stage, ethyl acetate was used, which forms two phases with DES, and an efficiency of 95% was obtained.

### 3.3. Distillation

Distillation is a process developed centuries ago in which differences in the boiling points of compounds selectively separate compounds. The more significant the difference in relative volatility, the greater the nonlinearity and the easier it is to separate the mixture. An essential aspect of achieving the separation is to know the equilibrium diagrams of the liquid (x)-vapor (Y) phases. Special care must be applied when working with an azeotrope mixture in which the mix behaves—at a specific composition and a constant boiling temperature—as if it were made up of a single component; that is, at the boiling temperature, the composition of the vapor phase is the same as in the liquid phase. In this way, the mixture cannot be separated [97].

Industrially, two types of distillation are used: batch and continuous. Batch distillation is the oldest method and has been used for centuries. The system consists of a bottom boiler with the feed to be separated and a rectifying column (plate or packed) superimposed on the boiler; both are connected to a condenser. In continuous distillation, the spread (liquid mixture) to be separated is fed to the column at one or more points along the column. The vapor moves up while, in the bottom, the liquid rich in heavy components is withdrawn from the column. The vapor that leaves the top of the column is condensed [98]. Part of the condensed liquid is recycled into the column while the rest is extracted as distillate.

In biotransformations, distillation is applied in various processes, for example, in the enzymatic synthesis of diacylglycerol (DAG) from glycerol and soybean oil using *t*-butanol as a solvent. DAGs are used as food additives for the formulation of low-fat foods. To recover the DAG product of interest, the researchers used vacuum distillation to remove t-butanol from the mixture [99] (Figure 8). A 98.7% conversion rate was successfully attained by employing a molar ratio of 6.23:1 for soybean oil to glycerol and utilizing a 40% *w*/*v* substrate dissolved in *t*-butanol. After the reaction had concluded, the concentration of DAG reached 48.5%. Further refinement through molecular distillation ultimately yielded an impressive purity level of 96.1%.

In chemoenzymatic synthesis of the biolubricant formed by fatty acid esters of hydroxy tetrahydrofuran, the distillation processes are essential to obtain the refined product. First, the hydration of linoleic acid takes place and then an epoxidation reaction. After epoxidation, vacuum distillation is used to obtain the epoxide that, together with the alcohol, will be esterified and catalyzed by the Novozym 435^®^ biocatalyst. Short-path distillation is then carried out, the condensate of which corresponds to the precursor ester for the biolubricant that is finally obtained. This process is detailed in Figure 9 [100].

### 3.4. Crystallization

Crystallization from solution is one of the most used unit operations in the chemical process industry. The formation and separation of crystal particles from solution involve periods of suspension and sedimentation. During these periods, fluid particles move and matter can change from liquid to solid [101,102].

Phase equilibrium and crystallization modes are the principles and techniques for developing a crystallization process. The development of phase equilibria considers the solubility of a component in a solvent depending on its concentration and temperature. Also, a strategy used is the application of an anti-solvent, which allows one to solubilize components that other solvents cannot, improving crystallization. Crystallization methods make use of supersaturation either by cooling or evaporation. The “precipitation”, supersaturation crystallization, is generated by adding a third component that induces a chemical reaction to produce the solute or reduce its solubility [101].

Various types of crystallizers exist, including unstirred and stirred vessels (See Figure 10). Unstirred vessels are the most straightforward equipment, where hot liquor is poured and cooled to room temperature in batch mode. In this mode of operation, local and transient variations in supersaturation occur, resulting in large, interlocked, agglomerated, and fine crystals that are difficult to separate (Figure 10A). In the case of agitated crystallizers, mass transfer in the system is improved, scale formation is reduced, supersaturation profiles are smoothed, crystals are suspended, a more uniform product is generated, and crystallization time is reduced (Figure 10B). Several alternatives exist for large-scale continuous crystal production, as detailed in Figure 10C.

*D*-Allulose is a low-calorie sugar used in processed foods to enhance flavor and appearance. However, producing *D*-Allulose from *D*-Glucose catalyzed by immobilized glucose isomerase will result in an additional by-product, decreasing yield and hindering the separation process. Fortunately, simulated moving bed chromatography (SMBC) followed by purification through crystallization is an efficient method for separating *D*-Allulose. This method has the advantages of low solvent consumption and high productivity and purity. After passing through the column, the *D*-Allulose solution is concentrated up to 3% using a nanofiltration membrane and then concentrated in a rotary evaporator under a vacuum at 65 °C until the sugar concentration reaches 70%. Anhydrous ethanol was added at a ratio of 1:4 (*v*/*v*), anhydrous ethanol to *D*-Allulose); subsequently, *D*-Allulose crystal seeds were added to start the crystallization process at a temperature of 4 °C for 96 h, allowing one to achieve 98% purity [103].

An interesting methodology to integrate crystallization in biotransformations is using bioreactors of membranes. An example is synthesizing (*S*)-1-(3-methoxyphenyl) ethylamine, a drug precursor against Alzheimer’s disease. A membrane bioreactor with two chambers is used to conduct the synthesis, wherein the reaction is first carried out. In the second chamber, the reaction products are concentrated and pass to a crystallization stage to obtain a product with 99.5% purity [104].

### 3.5. Chromatography

Chromatography can fulfill two essential functions. The first is as part of a downstream purification process in synthesis reactions to separate mixture components. In biotransformations, chromatography is used in downstream processes to purify products. There are various types of chromatography, including paper chromatography, thin-layer chromatography, liquid chromatography, gas chromatography, supercritical fluid chromatography, etc. In the case of liquid chromatography, the sample is eluted to be separated through a stationary phase (packed column). According to the isolated compounds’ characteristics, the elution is carried out using mixtures of solvents and water or only solvents. The most critical aspects of chromatography are the manufacture of the column—the packing of which can be made of various materials, particle sizes, and pore sizes—and the mobile phases that can use different solvents and water in various proportions [105].

Chromatography is a fundamental tool in the production of chiral compounds. This is the case in the research carried out by Sun et al. 2020 on the production of *D*-Pantoic acid and additive ingredients used in the food, pharmaceutical, and other industries. The enzymatic kinetic solution of *D*-Pantoic acid shows apparent advantages in its low production cost, toxicity, and pollution. The objective of this work was the cloning of the Lactonohydrolases (Lacs) enzymes that catalyze the hydrolysis of *D*/*L*-pantolactone (*DL-PL*) in D-pantoic acid (*D*-PA) or *L*-pantoic acid (*L*-PA). The final reaction products were monitored using HPLC provided with a chiral column, and the mobile phase was composed of methanol and 0.1% formic acid. The enantiomeric excess reached 99% after 4 h of reaction [106].

### 3.6. Selection and Comparison of Separation Methods

The selection of appropriate separation techniques depends on the type of product that needs to be separated, its physical characteristics, and its degree of purity. Capital expenditures (CAPEXs) and operational expenses (OPEXs) are also crucial, as Figure 11 details. For example, chromatography is a good choice if a high-purity product is required and there are a high number of byproducts. However, the CAPEX and OPEX will be high, and the energy required will be considerable.

If the products have different polarities, liquid–liquid extraction could be a good alternative if the solvents can be reused and are environmentally friendly.

Distillation is one of the classic separation processes; despite the high energy cost, it should be one of the least used alternatives. However, if it must be used, it should be selected for products with very different boiling points and where possibilities exist for the recovery and reuse of the byproducts.

Filtration is an expensive separation methodology, from an energy point of view, with high investment and maintenance to replace the membranes once they are used. However, in specific processes in which the products of interest are labile to high-temperature conditions and solvents, it is an appropriate alternative to other separation strategies.

## 4. Conclusions

The correct combination of upstream and downstream processes facilitates the development of sustainable and productive bioprocesses. Upstream operations have a significant impact on downstream processes. For this reason, enzyme production, immobilization, solvent selection, and optimization techniques help to enhance the development of enzyme technology on an industrial scale. Various downstream processes constitute about 80% of the total cost of the industrial process. Therefore, they must be strategically selected according to the kind of product, simplicity, scale-up, CAPEX, OPEX, and environmental considerations. Finally, the upstream and downstream processes must combine several characteristics such as simplicity, sustainability, and economy.

## Figures and Tables

**Figure 1 pharmaceutics-16-00038-f001:**
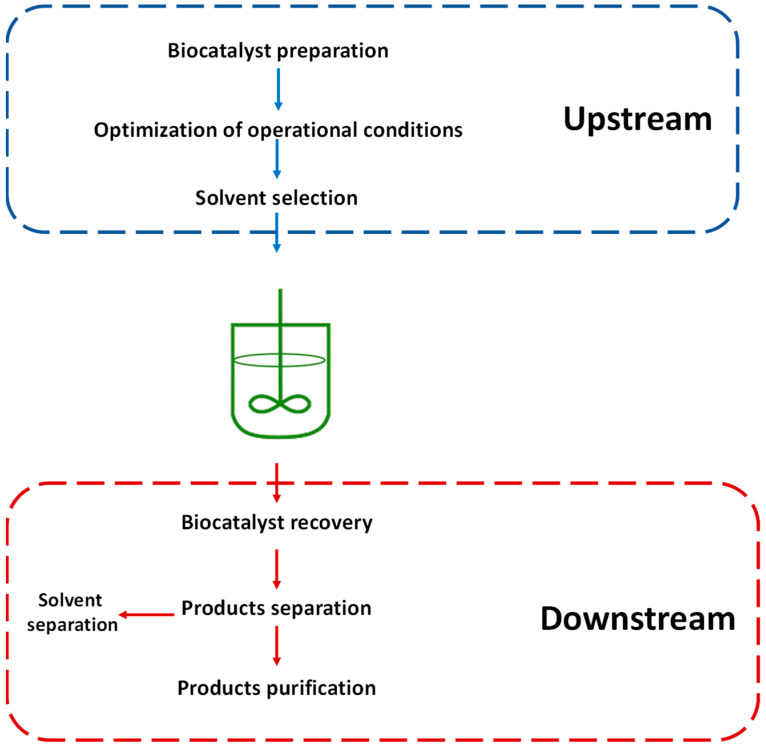
Upstream and downstream processes in enzyme technology.

**Figure 2 pharmaceutics-16-00038-f002:**
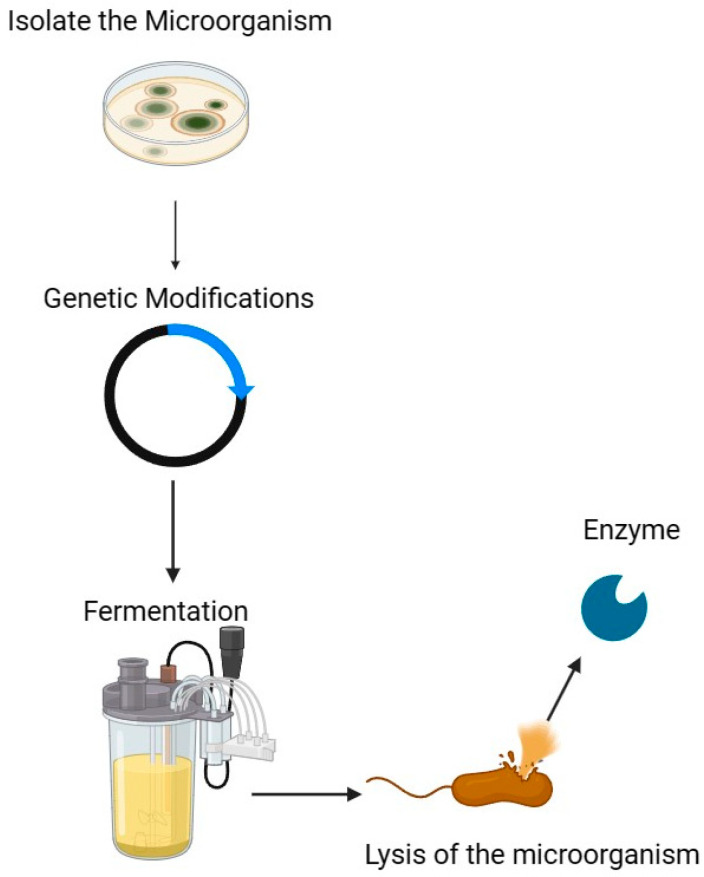
Enzyme production.

**Figure 3 pharmaceutics-16-00038-f003:**
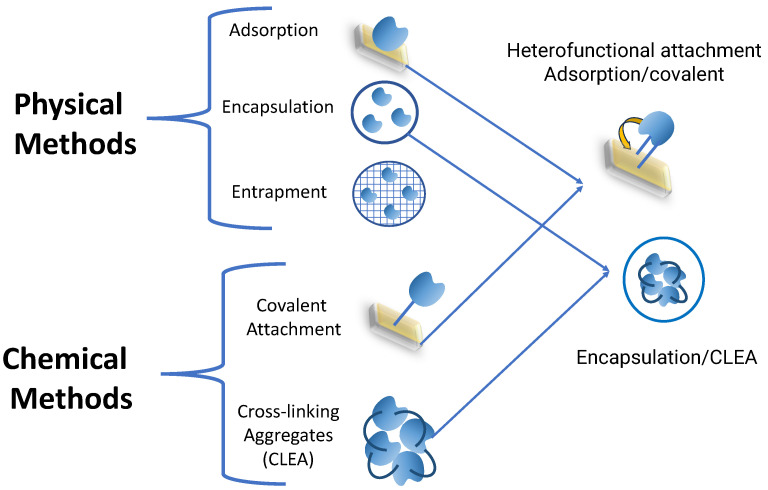
Types of enzyme immobilization.

**Figure 4 pharmaceutics-16-00038-f004:**
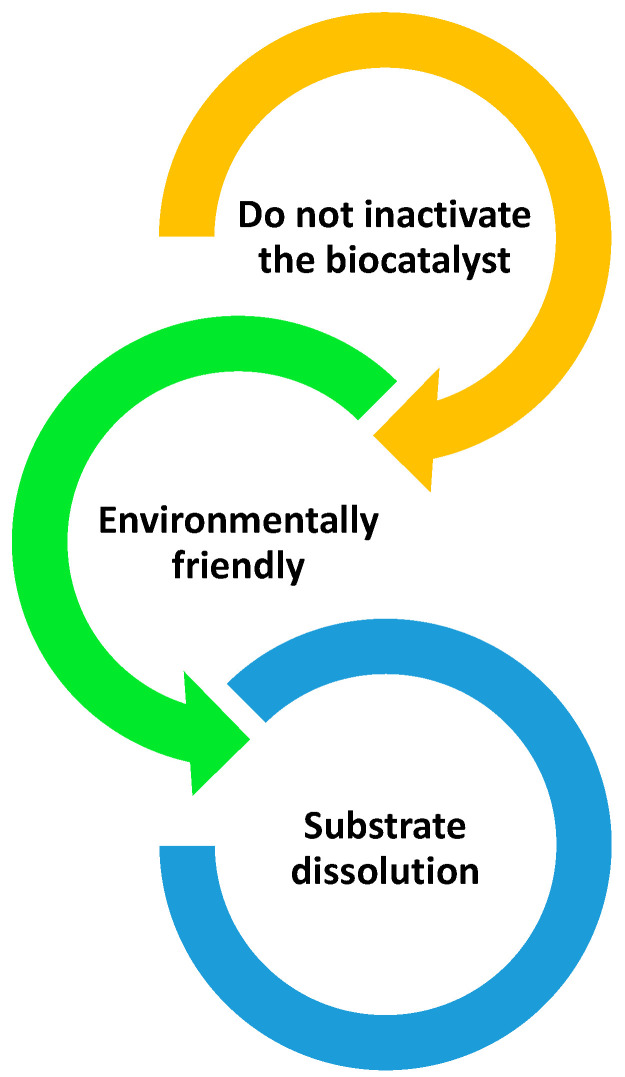
Main characteristics that solvents must have in enzymatic reactions.

**Figure 5 pharmaceutics-16-00038-f005:**
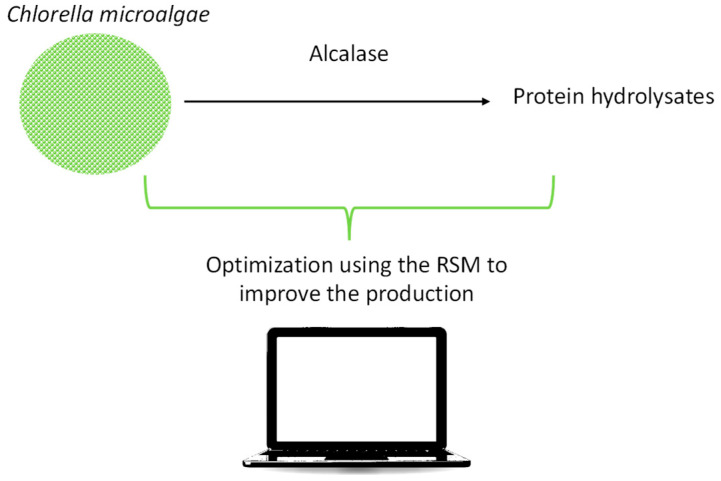
Optimization of the enzymatic hydrolysis of *Chlorella* microalgae to obtain protein hydrolysates.

**Figure 6 pharmaceutics-16-00038-f006:**
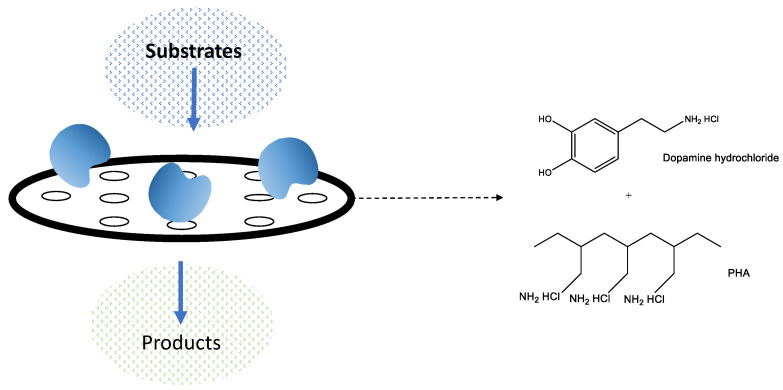
Membranes employed in fabricating membrane bioreactors enable concurrent separation of the biocatalyst and its products.

**Figure 7 pharmaceutics-16-00038-f007:**
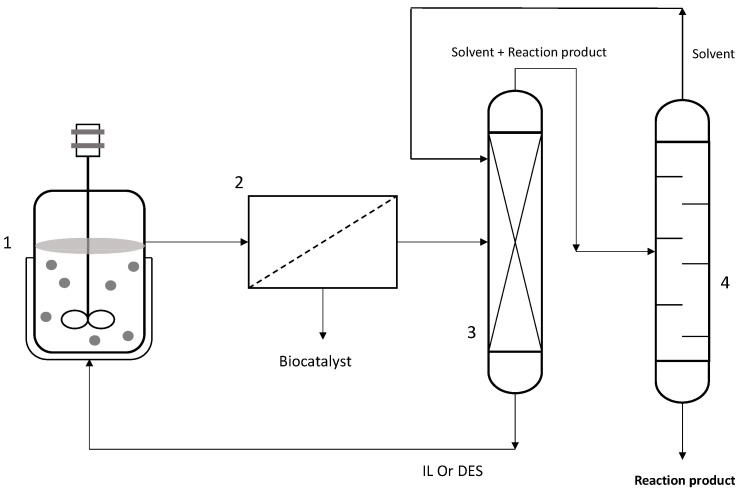
Downstream process with ILs and DESs. (1) Reaction system, (2) filtration, (3) liquid–liquid extraction, (4) distillation.

**Figure 8 pharmaceutics-16-00038-f008:**
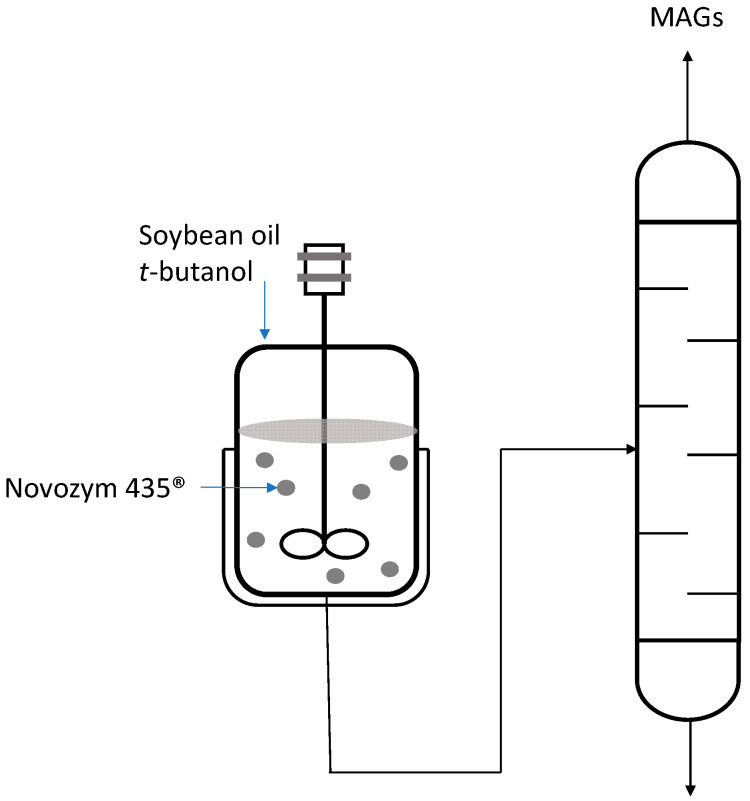
Purification of diacylglycerol produced by a glycerolysis reaction of soybean oil and glycerol catalyzed by the Novozym 435^®^ biocatalyst.

**Figure 9 pharmaceutics-16-00038-f009:**
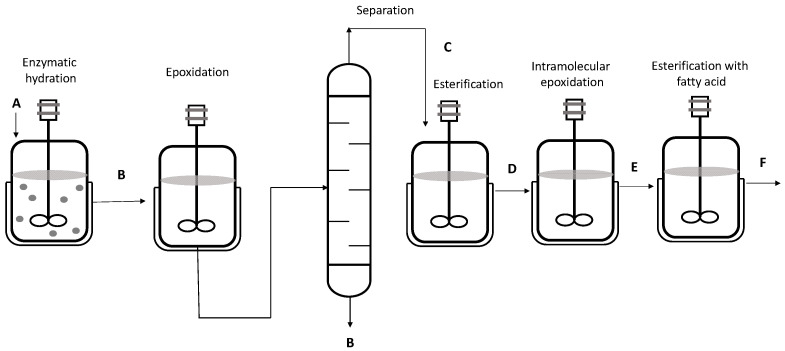
Linoleic-based hydroxytetrahydrofuran (HTHF) esters as biolubricant. (A) Linoleic acid, (B) 10-hydroxy-cis-12-octadecanoic acid, (C) 12,13-epoxy-10-hydroxy-octadecanoic acid, (D) 12,13-epoxy-10-hydroxy-octadecanoic acid ester, (E) 12,13-epoxy-12-hydroxy-octadecanoic acid ester, and (F) HTHF fatty acid esters.

**Figure 10 pharmaceutics-16-00038-f010:**
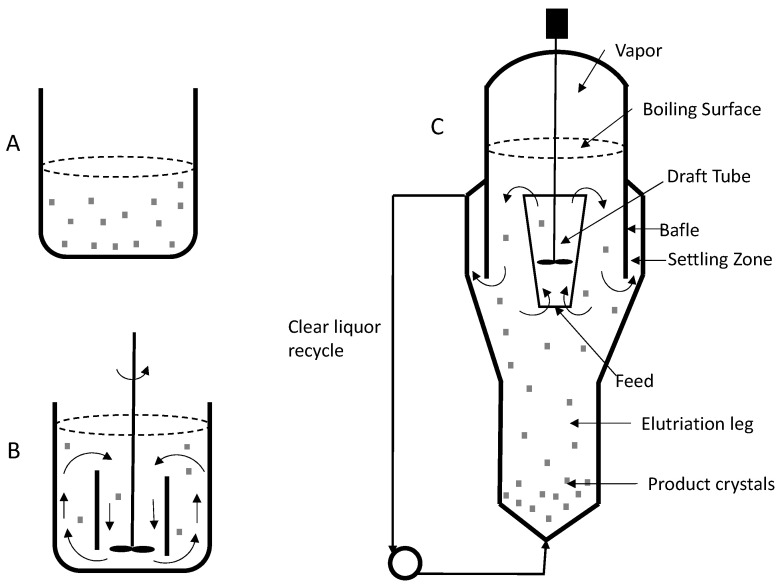
Crystallization equipment. (**A**) Unstirred vessel, (**B**) agitated vessel, (**C**) agitated crystallizers DTB (draft tube and baffle).

**Figure 11 pharmaceutics-16-00038-f011:**
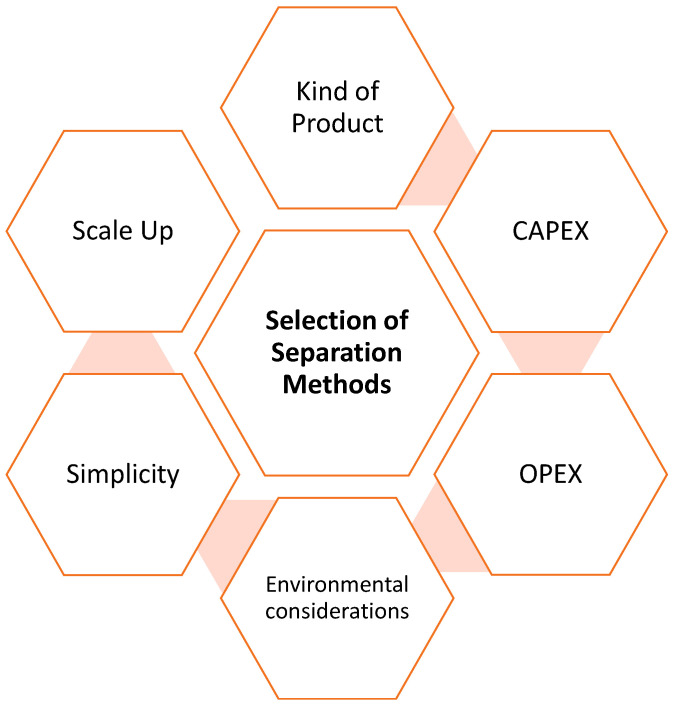
Considerations for the selection of separation methods.

## Data Availability

All information is in the references.

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
