# Peer review of "Upstream and Downstream Bioprocessing in Enzyme Technology"

_pharmaceutics, 2023, doi:10.3390/pharmaceutics16010038_

Round 1

Reviewer 1 Report

Comments and Suggestions for Authors

In this review, the authors presented recent examples of upstream and downstream processes related to enzyme biotechnology.

line 243 CLEAs has already been abbreviated above. Also note the abbreviation for ionic liquids (ILs) (lines 375 and 377). Please check the correctness of other abbreviations in the text.

lines 247-248 Please use italics for names of microorganisms.

line 317 in "and 16 U. [77].", please remove the extra period. Also check the spaces in the text.

line 335 Please use Greek beta instead of B in the name of the enzyme "B-galactosidase".

Line 386 Please give examples of ionic liquids and renewable sources for their production.

line 474 The D-Allulose have different spellings in the text, please correct.

The review clearly lacks several examples of combining upstream processes with downstream processes and a discussion of recommendations for combining these processes to ensure simplicity, sustanability and cost-effectiveness of the entire bioprocess.

Comments on the Quality of English Language

English language is good, no serious comments on it.

Author Response

In this review, the authors presented recent examples of upstream and downstream processes related to enzyme biotechnology.

line 243 CLEAs has already been abbreviated above. Also note the abbreviation for ionic liquids (ILs) (lines 375 and 377). Please check the correctness of other abbreviations in the text.

The reviewer is correct. The error was corrected. Please, see:

  • Page 1, line 42
  • Page 6, line 201
  • Page 19, line 218
  • Page 8, lines 250, 254, 255, 258, 260, 266, 273, and 276.
  • Page 9, lines 281, 291, 292, and 294.
  • Page 12, lines 389, 390, 391, 396, and 397.
  • Page 13, line 405
  • Page 14, lines 440, 441, and 442.

lines 247-248 Please use italics for names of microorganisms.

The mistake was corrected see page 8, line 259.

line 317 in "and 16 U. [77].", please remove the extra period. Also check the spaces in the text.

The mistake was corrected see page 10, line 330.

line 335 Please use Greek beta instead of B in the name of the enzyme "B-galactosidase".

The mistake was corrected see page 11, line 348.

Line 386 Please give examples of ionic liquids and renewable sources for their production.

The following examples were included with the reference: cholinium ethanoate, cholinium propanoate. Please see Page 12, line 399.

line 474 The D-Allulose have different spellings in the text, please correct.

The mistake was corrected see page 16, lines 496, 497, 500, 502, and 505.

The review clearly lacks several examples of combining upstream processes with downstream processes and a discussion of recommendations for combining these processes to ensure simplicity, sustanability and cost-effectiveness of the entire bioprocess.

The reviewer is correct. However, the work's structure has been organized to describe the upstream and downstream processes separately. For this reason, the word "combination" has been removed from the title.

Reviewer 2 Report

Comments and Suggestions for Authors

The review underscores the pivotal role of advanced recovery and purification techniques in shaping efficient biocatalytic processes during downstream operations. It gives due importance to enzyme immobilization, a crucial upstream process, exploring its impact on catalytic efficiency and enzyme stability. Although various immobilization methods are discussed, a comparative analysis or specific recommendations for optimal conditions would add depth. The review underscores the significance of solvent selection for biotransformations, acknowledging its influence on reaction rates and enzyme activity. However, a more detailed exploration of solvent selection criteria or examples of successful applications would enhance clarity.

Moving on to downstream techniquesfiltration, liquid-liquid extraction, distillation, crystallization, and chromatographyare explored for product separation and purification. The discussion on filtration methods, encompassing pressure and vacuum filters, provides valuable insights into their industrial applications. The application of nanofiltration in purifying enzymatically produced galactooligosaccharides serves as a practical case study. Liquid-liquid extraction, utilizing ionic liquids (ILs) and deep eutectic solvents (DES), is outlined for separating reaction products, offering a broad overview of their pros and cons.

The review delves into the role of distillation in diverse biotransformation processes, underscoring its significance in product recovery. Crystallization and chromatography are expounded as vital downstream operations, each exemplified with instances like D-allulose production and the chemoenzymatic synthesis of bio lubricants. However, critical evaluations or comparisons among these techniques are lacking, constraining the practical guidance provided. The conclusion, while summarizing key aspects, could benefit from a more elaborate connection with preceding discussions. While the review offers a comprehensive overview, incorporating specific case studies and comparative analyses would elevate its practical utility.

Comments on the Quality of English Language

.

Author Response

The review underscores the pivotal role of advanced recovery and purification techniques in shaping efficient biocatalytic processes during downstream operations. It gives due importance to enzyme immobilization, a crucial upstream process, exploring its impact on catalytic efficiency and enzyme stability. Although various immobilization methods are discussed, a comparative analysis or specific recommendations for optimal conditions would add depth.

The reviewer is correct. Added the following paragraph on page 8, lines 269-277.

“The selection of the immobilization methodology depends on the type of enzyme, simplicity, cost of the support, and the kind of reactor when a process is required to be scaled. For example, in the immobilization of lipases with industrial applications, absorption or heterofunctional methodologies (adsorption and covalent bonding) on methacrylate resins are a good option due to the high mechanical and chemical stability of the support for their application in batch and continuous packed bed bioreactors.

Immobilization techniques by encapsulation or entrapment give rise to biocatalysts with weak mechanical resistance, and their operation in batch bioreactors with mechanical stirrers constitutes an appropriate alternative for their scale-up.”

The review underscores the significance of solvent selection for biotransformations, acknowledging its influence on reaction rates and enzyme activity. However, a more detailed exploration of solvent selection criteria or examples of successful applications would enhance clarity.

As the reviewer suggested, a brief explanation was included regarding the selection of green solvents. Please, see page 8, lines 281-284.

 “In addition to the points noted in Figure 4, the solvent can also improve the enzyme activity to increase the reaction's conversion and yield. Depending on the polarity of the substrates, green solvents that can dissolve the substrates can be designed.”

Moving on to downstream techniques—filtration, liquid-liquid extraction, distillation, crystallization, and chromatography—are explored for product separation and purification. The discussion on filtration methods, encompassing pressure and vacuum filters, provides valuable insights into their industrial applications. The application of nanofiltration in purifying enzymatically produced galactooligosaccharides serves as a practical case study. Liquid-liquid extraction, utilizing ionic liquids (ILs) and deep eutectic solvents (DES), is outlined for separating reaction products, offering a broad overview of their pros and cons.

The review delves into the role of distillation in diverse biotransformation processes, underscoring its significance in product recovery. Crystallization and chromatography are expounded as vital downstream operations, each exemplified with instances like D-allulose production and the chemoenzymatic synthesis of bio lubricants. However, critical evaluations or comparisons among these techniques are lacking, constraining the practical guidance provided.

Section 3.6, "Selection and Comparison of Separation Methods," was incorporated, where the primary considerations for selecting separation methodologies are compared and detailed. Please see Page 17, lines 550- 568

The conclusion, while summarizing key aspects, could benefit from a more elaborate connection with preceding discussions.

The conclusions were enriched by including what was stated in section 3.6:

“For this reason, they must be strategically selected according to the kind of product, simplicity, scale-up, CAPEX, OPEX, and environmental considerations”

While the review offers a comprehensive overview, incorporating specific case studies and comparative analyses would elevate its practical utility.

The manuscript includes case studies such as the one on page 16, lines 508-519. Another example is the one on page 14, lines 452-460, among other examples mentioned in the manuscript.

Round 2

Reviewer 1 Report

Comments and Suggestions for Authors

The authors have substantially improved the text of the manuscript, in accordance with the reviewers' comments. I have no more significant comments to the revised manuscript and propose to accept it for publication in the journal.

Comments on the Quality of English Language

The English is good.